# Fission Yeast TORC2 Signaling Pathway Ensures Cell Proliferation under Glucose-Limited, Nitrogen-Replete Conditions

**DOI:** 10.3390/biom11101465

**Published:** 2021-10-06

**Authors:** Yusuke Toyoda, Shigeaki Saitoh

**Affiliations:** Institute of Life Science, Kurume University, Asahi-machi 67, Kurume, Fukuoka 830-0011, Japan; toyoda_yuusuke@kurume-u.ac.jp

**Keywords:** glucose limitation, nitrogen starvation, hexose transporter, TORC2, Gad8/AKT kinase, arrestin, ubiquitylation, endocytosis

## Abstract

Target of rapamycin (TOR) kinases form two distinct complexes, TORC1 and TORC2, which are evolutionarily conserved among eukaryotes. These complexes control intracellular biochemical processes in response to changes in extracellular nutrient conditions. Previous studies using the fission yeast, *Schizosaccharomyces pombe*, showed that the TORC2 signaling pathway, which is essential for cell proliferation under glucose-limited conditions, ensures cell-surface localization of a high-affinity hexose transporter, Ght5, by downregulating its endocytosis. The TORC2 signaling pathway retains Ght5 on the cell surface, depending on the presence of nitrogen sources in medium. Ght5 is transported to vacuoles upon nitrogen starvation. In this review, we discuss the molecular mechanisms underlying this regulation to cope with nutritional stress, a response which may be conserved from yeasts to mammals.

## 1. Introduction

Eukaryotic cells modulate intracellular biochemical processes in response to changes in availability of nutrients for proper growth, proliferation, and survival under a given extracellular microenvironment. While cells require nutrients as energy and substrates for growth and proliferation, nutritional conditions in the environment are not always favorable. Thus, cells need mechanisms to cope with nutritional stresses, such as starvation, for adaptation and survival. The TOR (Target of Rapamycin) kinases are critical for cellular responses to changes in nutritional conditions. As reviewed in the previous article by Tatebe and Shiozaki [1], the TOR kinases, which belong to the family of phosphatidylinositol 3-kinase related kinases, are evolutionally conserved among eukaryotes from yeasts to humans. TOR kinases form two distinct complexes, TORC1 (TOR Complex 1) and TORC2 (TOR Complex 2), which have different regulatory functions. While TORC1 and TORC2 share the same catalytic subunit (mTOR) in higher eukaryotes, including mammals, in yeasts these complexes contain a closely related but different protein as the catalytic subunit. In the fission yeast, *Schizosaccharomyces pombe*, TORC2 contains a TOR kinase encoded by the *tor1*^+^ gene as the catalytic subunit as well as TORC2-specific RICTOR and SIN1 proteins, which are encoded by the *ste20*^+^ and *sin1*^+^ genes, respectively [2,3]. *S. pombe* TORC1 contains a catalytic subunit, encoded by the *tor2*^+^ gene, and the TORC1-specific subunit RAPTOR, which is encoded by the *mip1*^+^ gene.

To perform their physiological functions, TORC1 and TORC2 phosphorylate and activate downstream effector kinases belonging to the AGC-family [4]. An AGC-family kinase phosphorylated by mammalian TORC2 (mTORC2) is AKT/protein kinase B (PKB). The C-terminal hydrophobic motif of AKT is phosphorylated by mTORC2, and the T-loop is phosphorylated by phosphoinositide-dependent protein kinase 1 (PDK1), which is recruited to membranes containing phosphatidylinositol-3,4,5-trisphosphate (PIP3). Phosphorylation at both sites is required for full activation of AKT in response to insulin and insulin-like growth factors. In fission yeast, an AGC-family kinase encoded by the *gad8*^+^ gene is proposed to be equivalent to mammalian AKT [5]. Like AKT, Gad8 kinase is phosphorylated and activated by TORC2 and a PDK1-like kinase, Ksg1 [6]. As fission yeast mutant cells defective in Gad8 largely share the phenotypes, including retarded cell cycle progression, with mutants defective in components of TORC2 [3,5,6], Gad8 appears to be the primary effector kinase in the fission yeast TORC2 signaling pathway.

Several groups, including ours, have reported that as in other organisms, the TORC2 signaling pathway in fission yeast regulates glucose metabolism and enables proper cell proliferation under glucose-limited conditions. Recently, we found that the *S. pombe* TORC2 signaling pathway inhibits endocytic vesicle trafficking of a high-affinity hexose transporter, Ght5, from the plasma membrane to vacuoles to ensure cell surface abundance of Ght5 and efficient glucose uptake under glucose-limited conditions [7]. Curiously, this TORC2-dependent inhibition of vacuolar trafficking of Ght5 is controlled in response to changes in the availability of nitrogen sources but not hexose. In this review, we discuss molecular mechanisms regulating vesicle trafficking of hexose transporters by the TORC2-Gad8/AKT signaling pathway. This mechanism appears to be essential for fission yeast cells to cope with nutritional stresses due to glucose and amino acid starvation. Importantly, mammalian AKT is reported to regulate endocytosis of hexose transporters in a manner similar to that in fission yeast [8], indicating that this mechanism is conserved during evolution of eukaryotes.

## 2. TORC2-AKT Is Required for Cellular Responses to Glucose and Nitrogen Starvation

While laboratory media for yeasts are normally supplied with high concentrations (2–3%, 111–168 mM) of glucose, fission yeast cells can grow and proliferate in medium containing only 0.08% (4.4 mM) glucose, which is equivalent to concentrations in normal human blood [9]. To proliferate under such glucose-limited conditions, *S. pombe* cells require TORC2 as well as AMP-activated protein kinase (AMPK) signaling pathways [10]. AMPK, which is activated upon decreased ATP production, regulates a wide range of cellular functions, including transcription, metabolism, cell growth, and autophagy, to generate ATP [11]. *S. pombe* mutant cells defective for a component of the TORC2 signaling pathway (Tor1/the TORC2 catalytic subunit, Ste20/RICTOR, Ksg1/PDK1 or Gad8/AKT) or the AMPK pathway (Ssp2/AMPK or Ssp1/Ca^2+^-calmodulin-dependent kinase that activates AMPK) fail to form colonies on glucose-limited solid media, while they proliferate and form colonies on glucose-rich (>2%) media. The high-affinity hexose transporter, Ght5, which is encoded by the *ght5*^+^ gene, is indispensable for cell proliferation under glucose-limited conditions. In wild-type cells, Ght5 is localized on the plasma membrane and mediates transport of glucose from the medium to the cytoplasm. While the AMPK signaling pathway is required for transcriptional upregulation of *ght5*^+^ upon glucose limitation, the TORC2 pathway is necessary for Ght5 localization on the plasma membrane [10,12]. Defects in the TORC2-Gad8/AKT signaling pathway cause abnormal accumulation of GFP, the degradation product of Ght5-GFP in vacuoles [10]. Consistently, the rate of glucose uptake is greatly reduced in TORC2-deficient mutant cells. Proliferation failure of mutant cells under glucose-limited conditions supposedly stems from this reduced glucose uptake. These observations indicate that the TORC2-Gad8/AKT signaling pathway is critical for the cellular response to glucose starvation.

Fission yeast genes for the TORC2-AKT signaling were originally identified as those required for cell cycle arrest at G1 phase upon nitrogen starvation [6,13,14,15]. Upon depletion of the nitrogen source (normally ammonium chloride) in the medium, wild-type fission yeast cells enter the sexual reproduction phase after a G1 cell cycle arrest. Haploid cells mate with those of the other mating type forming heterozygotic diploid cells, which undergo meiosis and subsequently form four spores. The TORC2 signaling pathway is required for induction of these sexual processes, and TORC2-deficiency causes sterility in fission yeast. These observations indicate that the TORC2 signaling pathway is also required for *S. pombe* cells to respond to nitrogen starvation. Consistent with this, TORC2 is involved in transcriptional regulation of a gene encoding an amino acid transporter protein upon nitrogen starvation [16].

Collectively, *S. pombe* TORC2-AKT is central to cellular responses in both nitrogen and glucose starvation, to which fission yeast cells respond differently. Further studies are required for a full understanding of molecular mechanisms regulating the activities and/or substrate specificities of TORC2 and AKT depending on concentrations of amino acids and glucose.

## 3. TORC2 Regulates α-arrestin for Persistence of Hexose Transporters on the Plasma Membrane

As mentioned in the previous section, fission yeast TORC2-Gad8/AKT is required for proper localization of the Ght5 hexose transporter on the cell surface. To reveal the mechanism underlying TORC2-dependent localization of Ght5 on the cell surface, Toyoda et al. screened for genomic suppressor mutations that rescue the growth deficiency of *S. pombe gad8* mutant cells in glucose-limited media [7]. In this screen, mutations in genes encoding components of the machinery for vesicular transport of proteins to vacuoles (i.e., components of ESCRT (Endosomal Sorting Complexes Required for Transport)) were isolated, suggesting that TORC2 acts in opposition to vesicle transport of the Ght5 hexose transporter from the surface to vacuoles.

Among mutations identified in the screening, *aly3* gene mutation was the strongest suppressor. Loss-of-function mutations in the *aly3^+^* gene greatly rescue growth deficiency under glucose-limited conditions and fully restore cell surface localization of the Ght5 transporter in *gad8* mutant cells. The *aly3*^+^ gene encodes a novel α-arrestin protein belonging to the arrestin superfamily. Arrestin superfamily proteins are evolutionarily conserved among eukaryotes and classified into four subfamilies: α-arrestins, β-arrestins, visual-arrestins, and Vps26-related proteins [17]. Aly3 is one of four conventional α-arrestins (Aly1, Aly2, Aly3, and Rod1) in *S. pombe*. Arrestin was originally identified as a protein terminating the signal from the light-activated photoreceptor, rhodopsin, in bovine eyes [18,19]. Arrestin proteins serve various functions, including as an adaptor connecting NEDD4-like E3 ubiquitin ligases to their substrate proteins on the plasma membrane in mammals. *S. pombe* Aly3 supposedly promotes association of Ght5 with a ubiquitin ligase, which has been proposed by Toyoda et al., but which remains to be identified, as well as its ubiquitylation [7] (Figure 1). Gad8/AKT appears to counter the action of Aly3, as mutations in TORC2-Gad8/AKT resulted in an increase of mono-ubiquitylated Ght5 in an Aly3-dependent manner. Mono-ubiquitylated Ght5 is supposedly transported to vacuoles via the multivesicular body (MVB) pathway, in which ESCRT protein complexes recognize ubiquitylated proteins in the endosome membrane and sort it to internalized vesicles to form MVBs [20]. MVBs fuse with vacuoles, which are equivalent to lysosomes, and proteins in internal vesicles of MVBs are subjected to proteolysis. The MVB outer membrane is fused and becomes a vacuolar membrane.

In the budding yeast, *Saccharomyces cerevisiae*, changes in the glucose concentration in the medium trigger internalization of hexose transporters from the cell surface to cytoplasmic vacuoles and subsequent proteolysis. Budding yeast cells express different hexose transporters on the cell surface depending on the concentrations and the types of hexoses in medium. For example, in the presence of high concentrations of glucose, low-affinity (but high-capacity) hexose transporters, such as Hxt1 and Hxt3, are expressed on the cell surface for glucose uptake, whereas high-affinity transporters, including Hxt6, are ubiquitylated and transported to vacuoles in a manner dependent on the arrestin, Rod1, and the E3 ubiquitin ligase, Rsp5 (Table 1) [21,22,23]. In contrast, in medium containing low concentrations of glucose or non-glucose carbon sources (e.g., galactose or lactate), Rod1 is inactivated via phosphorylation by the AMPK, Snf1, and consequently, high-affinity transporters are retained on the cell surface [22,24]. Under such glucose-limited conditions, low-affinity transporters undergo vacuolar transport [21,25]. Vacuolar transport and degradation of hexose transporters may facilitate exchange of the transporters on the cell surface; therefore, it appears reasonable that the change in glucose concentration triggers it.

Similarly, fission yeast cells use different types of hexose transporters depending on the glucose concentration in medium [10]. *S. pombe* has eight hexose transporters, designated Ght1 to Ght8 [26,27]. Under low-glucose conditions, Ght5 predominates on the plasma membrane, while Ght2 and Ght8 localize on the membrane under high-glucose conditions [10]. However, neither depletion nor repletion of glucose promotes trafficking of Ght5 to vacuoles, although the amount localized on the cell surface greatly changed in response to alteration in the glucose concentration due to the AMPK pathway-dependent transcriptional upregulation of the *ght5*^+^ gene in low glucose. Instead, depletion of the nitrogen source (ammonium chloride) triggers Aly3-dependent transport of Ght5 to vacuoles, regardless of glucose concentrations. As amino acid starvation due to an auxotroph mutation also causes vacuolar transport of Ght5, reduction of intracellular amino acid levels is supposedly the direct trigger of Ght5 transport to vacuoles. Upon amino acid starvation, TORC2 and Gad8/AKT, which inhibit Aly3-dependent endocytosis of Ght5, may be down-regulated by an unknown mechanism, and consequently, Ght5 is transported to vacuoles and degraded (Figure 1). It remains unclear why the availability of amino acids, not glucose, regulates vacuolar transport and subsequent proteolysis of the Ght5 glucose transporter. Intriguingly, *aly3* gene deletion mutant cells lose cell viability faster than wild-type fission yeast cells under nitrogen-depleted conditions, while cell viabilities of wild-type and mutant cells are similar in nitrogen-rich media. Aly3-dependent vacuolar transport and subsequent proteolysis of Ght5 (and possibly other cell surface proteins) may replenish intracellular amino acids essential to survive nitrogen starvation at the cost of glucose uptake ability. Reducing glucose uptake ability under nitrogen-starved conditions may be part of reestablishing metabolic homeostasis, in which fission yeast Aly3 serves an important function. It is noteworthy that the budding yeast TORC1 subunit, RAPTOR/Kog1, reportedly regulates metabolic homeostasis under glucose and amino acid limitation independently of TORC1 activity [28]. Therefore, components of both TORC1 and TORC2-Gad8/AKT signaling pathways may contribute to orchestration of metabolic homeostasis under nutrient limitation. Internalization of Ght5 may also secure a niche for other transporters on the plasma membrane. Upon nitrogen starvation, a fission yeast amino acid transporter, Aat1, which is confined to the Golgi apparatus via the actions of a β-arrestin, Any1, and a ubiquitin ligase, Pub1, under nitrogen-rich conditions, is transported to the cell surface [29,30]. The TORC2 signaling pathway in fission yeast may therefore optimize amounts of hexose and amino acid transporters expressed on the cell surface depending on amounts of available nutrients.

The above-mentioned results of Toyoda et al. imply that activity of the TORC2-Gad8/AKT pathway is regulated by abundance of nitrogen sources [7]. However, Hatano et al. suggest that activities of TORC2 and Gad8/AKT, which were measured as the level of the Gad8 protein phosphorylated at serine 546, are not affected by the nitrogen concentration in medium [31]. Although further studies are required to explain this apparent discrepancy, the substrate specificities of TORC2 and Gad8/AKT, rather than total kinase activities, are speculated to be regulated by nitrogen concentrations [7].

A schematic model showing how TORC2-AKT ensures cell-surface localization of the high affinity hexose transporter Ght5 and reflects our recent results [7]. In the presence of rich nitrogen sources in the medium, regardless of glucose levels, the TORC2-Gad8/AKT pathway inhibits Aly3-dependent ubiquitylation of Ght5, presumably by phosphorylation of Aly3 at serine 460, which is localized in the sequence matching a consensus sequence for AKT phosphorylation [32]. As a result, Aly3-dependent ubiquitylation of Ght5 is blocked, and Ght5 is preferentially localized on the cell surface under nitrogen-rich conditions, regardless of glucose levels. In either wild-type cells cultured under nitrogen-starved conditions or mutant cells defective in the TORC2-Gad8 pathway, Aly3 may be dephosphorylated and activated to stimulate ubiquitylation of Ght5 mediated by an unidentified E3 ubiquitin ligase of the NEDD4 family. Pub1, Pub2, and Pub3, the NEDD4 family ubiquitin ligases in *S. pombe*, may be responsible for ubiquitylation of Ght5. Once endocytosed, ubiquitylated Ght5 is recognized and internalized within MVB in intraluminal vesicles by the ESCRT, followed by the transport of such a species of Ght5 to vacuoles, in which the protein is degraded into amino acids. The regenerated amino acids may contribute to increased activity of the TORC2-Gad8/AKT pathway and/or cell survival under low-glucose, nitrogen-starved conditions.

## 4. Mammalian AKT also Regulates Arrestin-Mediated Internalization of Hexose Transporters on the Cell Membrane

As discussed above, arrestin-mediated transport of hexose transporters to vacuoles is regulated by the Gad8/AKT kinase in fission yeast, while it is regulated by the AMPK pathway in budding yeast. Similar to fission yeast, arrestin-mediated internalization of hexose transporters is reportedly regulated by AKT in mammalian cells. Mammalian thioredoxin-interacting protein (TXNIP) belongs to the α-arrestin family [17,33,34]. An important function of TXNIP, which has multiple functions, is regulation of cellular glucose uptake via glucose transporter proteins (GLUT1—4) in the plasma membrane [8,35]. TXNIP mediates physical interaction between hexose transporter molecules and clathrin protein, which participates in formation of clathrin-coated endocytic vesicles for endocytosis [36]. Importantly, in response to stimulation by insulin and insulin-like growth factors, TXNIP is phosphorylated at serine 308 by AKT [8]. This phosphorylation leads to dissociation of TXNIP from glucose transporter molecules and consequently prevents them from being endocytosed. Thus, regulatory mechanisms for internalization of hexose transporters through phosphorylation of arrestin by AKT/Gad8 appears to be conserved evolutionarily from fission yeast to mammals (Table 1). In mammalian cells stimulated by growth factors, in which energy demand is supposedly raised for cell growth and division, AKT may increase the glucose uptake rate by inhibiting internalization of hexose transporters as well as by promoting translocation of GLUT4 from cytoplasmic storage vesicles to the plasma membrane [8,37]. On the other hand, in fission yeast cells under nitrogen-rich conditions, Gad8/AKT may block unnecessary protein degradation for amino acid replenishment. It should be noted that TXNIP is also phosphorylated by AMPK at serine 308 upon acute glucose limitation [35]. In mammalian cells, therefore, both result in phosphorylation of TXNIP at serine 308, but the signaling pathways leading to phosphorylation are different.

## 5. Concluding Remarks

In this review, we have discussed how *S. pombe* TORC2-Gad8/AKT ensures glucose uptake and cell proliferation under glucose-limited conditions. As long as nitrogen sources are present in the medium, *S. pombe* TORC2-Gad8/AKT inactivates the α-arrestin, Aly3, which is essential for vesicle transport of the Ght5 hexose transporter from the plasma membrane to cytoplasmic vacuoles. Similarly, in cells of mammals, including humans, internalization of hexose transporters from the cell surface to endocytic vesicles is controlled by AKT-dependent phosphorylation of the α-arrestin, TXNIP. Thus, inactivation of an α-arrestin by phosphorylation appears to be an evolutionarily conserved function of TORC2-AKT, although it remains to be determined whether α-arrestins are regulated via the TORC2-AKT pathway in other model organisms. Further studies are required to uncover molecular mechanisms regulating AKT-dependent phosphorylation of arrestins in response to nutritional/growth hormone stimuli.

## Figures and Tables

**Figure 1 biomolecules-11-01465-f001:**
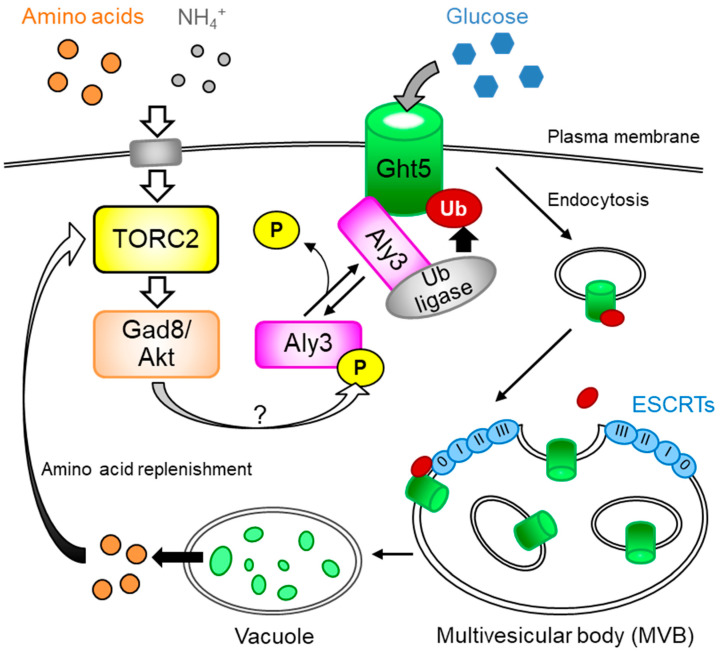
The TORC2-Gad8/AKT signaling pathway ensures cell-surface localization of the *S. pombe* hexose transporter, Ght5.

**Table 1 biomolecules-11-01465-t001:** Upstream stimuli and signaling cascades that control subcellular localization of hexose transporters in model organisms.

	*S. pombe*	Mammals	*S. cerevisiae* ^(^ ^1)^
Upstream stimuli	Nitrogen starvation	Glucose starvation, insulin-like growth factors	Glucose starvation
Signaling pathways	TORC2, Gad8/AKT	AMPK, TORC2, AKT	AMPK, PKA
Arrestins	Aly3	TXNIP	Rod1, Rog3, Csr2
Factors binding to arrestins	Ub ligase (Pub1/2/3?)	Clathrin, ITCH Ub ligase	Rsp5 Ub ligase
Targeted hexose transporters	Ght5	GLUT1-4	Hxt2, Hxt4, Hxt6, Hxt7, Hxt1, Hxt3

(1) See O’Donnell et al. (2019) [23] for details.

## Data Availability

No new data were created or analyzed in this study. Data sharing is not applicable to this article.

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
