# Peer review of "Fission Yeast TORC2 Signaling Pathway Ensures Cell Proliferation under Glucose-Limited, Nitrogen-Replete Conditions"

_biomolecules, 2021, doi:10.3390/biom11101465_

Round 1

Reviewer 1 Report

An interesting mini-review on the topic. Some editing is required. I make those suggestions I picked out while reading.

Line 29. Should also formally define TORC2 (TOR Complex 2)

line 35. I was expecting an equivalent brief description of TORC1

line 40. Define PDK1 kinase (PDPK1?). Is it also regulated? A sentence to contextualise is required.

Line 68 Define AMPK (you later do in ln 137). It again requires a brief explanation/context for non specialists.

Line 69. presumably ...PDK1 and Gad8… should be ‘or’ not ‘and’

line 80. ‘signalling was originally’ should be ‘signalling were originally’

line 85. undergo meiosis and subsequently form four spores.

Line 93. for a full understanding

line 103. In this screen…. (not screening)

line 105. ‘counteracts’ is the wrong word. TORC2 acts in opposition to vesicle transport.

Line 118. What do you mean by ‘supposedly’? Either it does (then reference) has been proposed to (then state and reference), or you are suggesting here for the first time?

Line 142. I suggest you start a new paragraph to describe the fission yeast situation. I also suggest you include a little more detail on the types of hexose transporters and add some references for the various observations discussed.

Line 154. ‘may be somehow downregulated’ A strange and vague statement. Please clarify what you mean, down regulated by an unknown mechanism?

Line 170. Cite reference number for Hatano et al.

Line 171. ‘...546th serine residue’. This is ambiguous as it implies not residue 546, but the 546th serine in a string of amino acids, which would not be residue 546 (unless the whole protein was simply made of serines’. Use the more established notation ‘phosphorylated at serine 546’ (or phosphorylated at S546). Similar criticism lines: 183, 224

line 186. Unidentified E3 ligase? You go on to identify it as either pub1, 2 or 3). Please clarify.

Line 206 ‘...TXNIP, which have multiple...’. change have for has.

Line 228. glucose-limited conditions. This is not correct? You mainly discuss the changes induced by nitrogen limiting conditions. A similar criticism could be made of the title. I suggest you consider making some changes to reflect this. i.e. ‘Fission yeast TORC signalling regulates sugar transport under nutrient depletion’

Author Response

Reviewer 1

Comments and Suggestions for Authors

An interesting mini-review on the topic. Some editing is required. I make those suggestions I picked out while reading.

              We thank the reviewer for the careful assessment and constructive criticisms of our manuscript (ms), which have helped to make the revised sections much clearer. Please find our point-by-point replies below, in blue. The English in the revised ms was also proofread by a native-speaking researcher. The ms has been revised using “Track Changes.”

Line 29. Should also formally define TORC2 (TOR Complex 2)

              TORC2 is formally defined at the suggested place (Line 32) in the revised ms.

line 35. I was expecting an equivalent brief description of TORC1

              A short description of the subunits constituting TORC1 has been added to the revised ms (Line 38).

line 40. Define PDK1 kinase (PDPK1?). Is it also regulated? A sentence to contextualise is required.

              In the revised ms, PDK1 is defined where it first appears (Line 45). A short description of the spatial regulation of PDK1 by phosphatidylinositol-3,4,5-trisphosphate (PIP3) has also been added.  

Line 68 Define AMPK (you later do in ln 137). It again requires a brief explanation/context for non specialists.

              In the revised ms, AMPK is defined where it first appears (Line 76). A brief explanation of AMPK has also been added.

Line 69. presumably ...PDK1 and Gad8… should be ‘or’ not ‘and’

              The reviewer is correct. (Line 81).

line 80. ‘signalling was originally’ should be ‘signalling were originally’

              The corresponding part is corrected (Line 97).

line 85. undergo meiosis and subsequently form four spores.

              This passage has been modified as suggested (Line 102).

Line 93. for a full understanding

              This has been corrected as indicated (Line 111).

line 103. In this screen…. (not screening)

              The wording has been modified as suggested (Line 121).

line 105. ‘counteracts’ is the wrong word. TORC2 acts in opposition to vesicle transport.

              This word has been replaced (Line 124).

Line 118. What do you mean by ‘supposedly’? Either it does (then reference) has been proposed to (then state and reference), or you are suggesting here for the first time?

              In the revised ms, we clarified the confusing sentence (S. pombe Aly3 supposedly promotes association of Ght5 with a ubiquitin ligase) by indicating that this is a hypothesis proposed in our recent publication. The reference has been added (Line 140).

Line 142. I suggest you start a new paragraph to describe the fission yeast situation. I also suggest you include a little more detail on the types of hexose transporters and add some references for the various observations discussed.

              In the revised ms, a new paragraph has been started to describe regulation of fission yeast hexose transporters (starting from Line 165).

              We have added a description of the types of S. pombe hexose transporters, “S. pombe has eight hexose transporters, designated Ght1 to Ght8 [26,27]. Under low-glucose conditions, Ght5 predominates on the plasma membrane, while Ght2 and Ght8 localize on the membrane under high-glucose conditions[10].” (Lines 166 to 169)

Line 154. ‘may be somehow downregulated’ A strange and vague statement. Please clarify what you mean, down regulated by an unknown mechanism?

              The indicated phrase of the revised ms has now been modified as suggested, “which inhibit Aly3-dependent endocytosis of Ght5, may be downregulated by an unknown mechanism and…” (Line 178).

Line 170. Cite reference number for Hatano et al.

              At the end of the indicated sentence of the revised ms, the reference number for Hatano et al. is now shown (Line 205).

Line 171. ‘...546th serine residue’. This is ambiguous as it implies not residue 546, but the 546th serine in a string of amino acids, which would not be residue 546 (unless the whole protein was simply made of serines’. Use the more established notation ‘phosphorylated at serine 546’ (or phosphorylated at S546). Similar criticism lines: 183, 224

              The corresponding parts of the revised manuscript have been modified as suggested (Lines 204, 218, 264)

line 186. Unidentified E3 ligase? You go on to identify it as either pub1, 2 or 3). Please clarify.

              In the revised ms, we clarified that the E3 ligase responsible for Ght5 ubiquitylation has not been determined yet, but one or more of the NEDD4 family Ub ligases, Pub1, Pub2 and Pub3, may be responsible for Ght5 ubiquitylation (Line 225).

Line 206 ‘...TXNIP, which have multiple...’. change have for has.

              The verb has been corrected as indicated (Line 244).

Line 228. glucose-limited conditions. This is not correct? You mainly discuss the changes induced by nitrogen limiting conditions. A similar criticism could be made of the title. I suggest you consider making some changes to reflect this. i.e. ‘Fission yeast TORC signalling regulates sugar transport under nutrient depletion’

              We have added a phrase “As long as nitrogen sources are present in the medium” (Line 267) to the revised ms to clarify what we discuss in the text.

             We have modified the title of the revised ms as “Fission yeast TORC2 signaling pathway ensures cell proliferation under glucose-limited, nitrogen-replete conditions” (Lines 2-3), to clarify what we discuss in the text.   

Reviewer 2 Report

In the mini-review authors summarized recent findings showing that endocytosis of Ght5, high affinity glucose transporter of fission yeast, is upregulated by downregulation of TORC2 Gad8/ACT signaling pathway in response to nitrogen starvation but regardless of glucose availability. Moreover, authors discussed the results and compared with the regulation of hexose transporters in human. The review is interesting but needs improvement.

Interesting message that cell surface abundance of glucose transporter is regulated in response to nitrogen sources to balance the nutrients needs upon nutrient limitation is missing in the title and abstract.

There is a recent very interesting paper by Rashida at al., 2021, Sci. Adv. 7: eabe5544 about the metabolic balance upon nutrient limitation which may be useful to enrich the interpretation of the results in L154 and below.

Unify the order, TORC2-AKT/Gad8 or TORC2-Gad8/AKT, in whole text.

Figure 1

Double line should be described as plasma membrane and should be elongated on the left. Low glucose on the top. High affinity hexose transporter, in the legend.

Table 1.

Content is not in agreement with the text. L202 Gad8/AKT in fission yeast but AKT is missing in the column for S. pombe. L211 by AKT but AKT missing in the column Mammals.

Abstract

L12, downregulating (indicate direction)

Introduction

L51, endocytic vesicle trafficking

L52, to the vacuoles to ensure cell surface abundance of Ght5 and efficient glucose uptake

L56, regulating vesicle trafficking of hexose transporters by the TORC2- Gad8/AKT.

L60, reference is missing

Chapter 1

L63, title is not clear to me. How glucose affects TORC2 and Ght5? Authors have to be very precise. Some information is missing in the chapter, how gene encoding Ght5 or Ght5 protein is regulated by glucose with the involvement of TORC2 or the title must be corrected.

L67, is glucose measured in the blood or serum?

L73, mediates transport of glucose

L75, statement “abnormal accumulation of Ght5 in the cytoplasm” is not correct. Rather, GFP, the degradation product of Ght5-GFP accumulates in the vacuoles, not the intact transmembrane protein. In Toyoda et al., 2021, Fig 1C and 2A, the diffused signal can be observed in the vacuole interior, not in vacuole membranes. That is in agreement with MVB sorting of Ght5 to internal vesicles.

L89, transcriptional regulation of a gene encoding amino acid transporter protein.

Chapter 2

L103, mutations in genes encoding components of the machinery for vesicular transport of proteins to the vacuoles.

L104 ESCORT abbreviation first appears here and must be explained as is in L123. Explanation must be removed in L123.

L115, add information, what organism?

L117 on the plasma membrane in mammals.

L123, remove all in brackets

L124, in the endosome membrane and select/sort it to internalized vesicles to form MVB

L125, proteins in internal vesicles of MVB are subjected to proteolysis. MVB outer membrane is fused and become a vacuolar membrane.

L140, vacuolar transport and degradation.

L146, it is contradiction, have to be added: in response to alteration in the glucose concentration due to the transcriptional regulation of the gene encoding Ght5. This have to be explained clearly to the reader, how glucose regulates transcription of ght5, what is known, which signaling pathway is involved etc. Otherwise is a mess: L146, greatly changed in response to glucose and LL148 regardless of glucose concentration.

L169, abundance of nitrogen sources

L181 of rich nitrogen sources

L183, serine460 residue which is localized in the sequence matching a consensus

L184Aly-dependent ubiquitilation of Ght5 is blocked and Ght5 is preferentially localized

L187,  nitrogen rich or starved conditions?

L187, to stimulate

L188, ligase of NEDD4 family

L189, recognized and internalized within MVB in intraluminal vesicles

MVB abbreviation should be explained only once, when first appears.

L192, to increase activity

Chapter 3

L205, (TXNIP) belongs to. Simpler and less words is better

L207 in the plasma membrane

L209 clathrin coated endocytic vesicles.

L216. Too long sentence, 6 lines, split.

L224, both result in phosphorylation of.. serine 308 residue. Simpler and shorter is better.

Concluding remarks

L232, to the endocytic vesicles

L234, remains to be determined

References

Be sure that Schizosaccharomyces pombe is in italics.

Author Response

Reviewer 2

Comments and Suggestions for Authors

In the mini-review authors summarized recent findings showing that endocytosis of Ght5, high affinity glucose transporter of fission yeast, is upregulated by downregulation of TORC2 Gad8/ACT signaling pathway in response to nitrogen starvation but regardless of glucose availability. Moreover, authors discussed the results and compared with the regulation of hexose transporters in human. The review is interesting but needs improvement.

              We thank the reviewer for the careful assessment and helpful comments, and we are pleased that he or she appreciates the significance of our article. As detailed below, in blue, we have responded to the reviewer’s comments in the revised manuscript (ms) by clarifying the text, the figure and the table. The English in the revised ms was also proofread by a native-speaking researcher. Modified parts of the revised ms have been corrected using “Track Changes.”

Interesting message that cell surface abundance of glucose transporter is regulated in response to nitrogen sources to balance the nutrients needs upon nutrient limitation is missing in the title and abstract.

              To clarify what we discuss in the text, we modified the title and added a sentence to the abstract of the revised ms as follows.

Title: Fission yeast TORC2 signaling pathway ensures cell proliferation under glucose-limited, nitrogen-replete conditions (Lines 2-3)

Abstract: The TORC2 signaling pathway retains Ght5 on the cell surface, depending on the presence of nitrogen sources in medium. Ght5 is transported to vacuoles upon nitrogen starvation. (Lines 13-15)

There is a recent very interesting paper by Rashida et al., 2021, Sci. Adv. 7: eabe5544 about the metabolic balance upon nutrient limitation which may be useful to enrich the interpretation of the results in L154 and below.

              We have added the following sentences to the revised ms, “Reducing glucose uptake ability under nitrogen-starved conditions may be part of reestablishing metabolic homeostasis, in which fission yeast Aly3 serves an important function. It is noteworthy that the budding yeast TORC1 subunit, RAPTOR/Kog1, reportedly regulates metabolic homeostasis under glucose and amino acid limitation independently of TORC1 activity [28]. Therefore, components of both TORC1 and TORC2-Gad8/AKT signaling pathways may contribute to orchestration of metabolic homeostasis under nutrient limitation.” (Lines 187-193)

Unify the order, TORC2-AKT/Gad8 or TORC2-Gad8/AKT, in whole text.

              The expression “TORC2-Gad8/AKT” is now used throughout the revised ms.

Figure 1

Double line should be described as plasma membrane and should be elongated on the left. Low glucose on the top. High affinity hexose transporter, in the legend.

              In the revised Figure 1, the double line is designated as “Plasma membrane” and is elongated to the left. Because TORC2-Gad8/AKT inhibition of Aly3-mediated endocytosis of Ght5 occurs regardless of glucose levels in the medium, we have added the phrase “regardless of glucose levels” (Line 216) to the revised legend to clarify this point. Also, the phrase “the high affinity hexose transporter” (Line 214) has been added to the revised legend.

Table 1.

Content is not in agreement with the text. L202 Gad8/AKT in fission yeast but AKT is missing in the column for S. pombe. L211 by AKT but AKT missing in the column Mammals.

              In the revised Table 1, “Gad8/AKT” and “AKT” have been added to the columns for S. pombe and Mammals, respectively.  

Abstract

L12, downregulating (indicate direction)

              The corresponding part of the revised ms has been modified as suggested, “by downregulating its endocytosis.” (Line 13)

Introduction

L51, endocytic vesicle trafficking

              This sentence has been modified in the revised ms as suggested, “the S. pombe TORC2 signaling pathway inhibits endocytic vesicle trafficking…” (Line 58)

L52, to the vacuoles to ensure cell surface abundance of Ght5 and efficient glucose uptake

              This passage now reads, “from the plasma membrane to vacuoles to ensure cell surface abundance of Ght5 and efficient glucose uptake...” (Line 59)

L56, regulating vesicle trafficking of hexose transporters by the TORC2- Gad8/AKT.

              We have edited this sentence as suggested, “regulating vesicle trafficking of hexose transporters by the TORC2-Gad8/AKT signaling pathway.” (Line 64)

L60, reference is missing

              The reference has been added to Line 68 of the revised ms.

Chapter 1

L63, title is not clear to me. How glucose affects TORC2 and Ght5? Authors have to be very precise. Some information is missing in the chapter, how gene encoding Ght5 or Ght5 protein is regulated by glucose with the involvement of TORC2 or the title must be corrected.

              In the revised ms, a description about regulation of ght5+ gene transcription in response to glucose levels has been added to clarify how TORC2-Gad8/AKT and AMPK contribute to cellular responses to glucose limitation, “The high-affinity hexose transporter, Ght5, which is encoded by the ght5+ gene, is indispensable for cell proliferation under glucose-limited conditions. In wild-type cells, Ght5 is localized on the plasma membrane and mediates transport of glucose from the medium to the cytoplasm. While the AMPK signaling pathway is required for transcriptional upregulation of ght5+ upon glucose limitation, the TORC2 pathway is necessary for Ght5 localization on the plasma membrane [10,12].” (Lines 84-89)

L67, is glucose measured in the blood or serum?

              Because we refer to the glucose level measured in the blood (not serum), this sentence (Line 75) was not changed in the revised ms. 

L73, mediates transport of glucose

              This sentence in the revised ms has been edited as suggested. (Line 86)

L75, statement “abnormal accumulation of Ght5 in the cytoplasm” is not correct. Rather, GFP, the degradation product of Ght5-GFP accumulates in the vacuoles, not the intact transmembrane protein. In Toyoda et al., 2021, Fig 1C and 2A, the diffused signal can be observed in the vacuole interior, not in vacuole membranes. That is in agreement with MVB sorting of Ght5 to internal vesicles.

              The cited phrase of the revised ms has been reworded as suggested, “cause abnormal accumulation of GFP, the degradation product of Ght5-GFP in vacuoles [10].” (Lines 92)

L89, transcriptional regulation of a gene encoding amino acid transporter protein.

              This has been changed as suggested. (Line 107)

Chapter 2

L103, mutations in genes encoding components of the machinery for vesicular transport of proteins to the vacuoles.

              This change has been made, as suggested. (Lines 122)

L104 ESCORT abbreviation first appears here and must be explained as is in L123. Explanation must be removed in L123.

              The ESCRT abbreviation has been moved from Line 123 in the original ms to Line 123 in the revised ms, where “ESCRT” first appears.

L115, add information, what organism?

              The name of the organism (bovine eyes) has been added (Line 135).

L117 on the plasma membrane in mammals.

              This has been modified as suggested. (Line 138)

L123, remove all in brackets

              The ESCRT abbreviation has been moved from Line 123 in the original ms to Line 123 in the revised ms, where “ESCRT” first appears.

L124, in the endosome membrane and select/sort it to internalized vesicles to form MVB

              We have done as the reviewer suggested. (Line 145)

L125, proteins in internal vesicles of MVB are subjected to proteolysis. MVB outer membrane is fused and become a vacuolar membrane.

              This has been rewritten as suggested. (Line 147)

L140, vacuolar transport and degradation.

              Modified as suggested. (Line 162)

L146, it is contradiction, have to be added: in response to alteration in the glucose concentration due to the transcriptional regulation of the gene encoding Ght5. This have to be explained clearly to the reader, how glucose regulates transcription of ght5, what is known, which signaling pathway is involved etc. Otherwise is a mess: L146, greatly changed in response to glucose and LL148 regardless of glucose concentration.

              In the revised ms, the indicated part is modified essentially as suggested. It now reads, “in response to alteration in the glucose concentration due to the AMPK pathway-dependent transcriptional upregulation of the ght5+ gene in low glucose.” (Lines 171-173)

L169, abundance of nitrogen sources

              The corresponding part of the revised ms is modified as suggested. (Line 202)

L181 of rich nitrogen sources

              This phrase has been revised as suggested. (Line 216)

L183, serine460 residue which is localized in the sequence matching a consensus

              We have used the wording recommended by the reviewer. (Line 218)

L184Aly-dependent ubiquitilation of Ght5 is blocked and Ght5 is preferentially localized

              The corresponding part of the revised ms is modified as suggested. (Lines 219-220)

L187,  nitrogen rich or starved conditions?

              In the original ms, we already described conditions in which Aly3 may be dephosphorylated and activated, as follows, “In either wild-type cells cultured under nitrogen-starved conditions or mutant cells defective in the TORC2-Gad8 pathway, Aly3 may be dephosphorylated and activated, …” (Line 222-223 of the revised ms). Therefore, we do not think that the nutrient conditions need to be mentioned again.

L187, to stimulate

              Done. (Line 224)

L188, ligase of NEDD4 family

              Modified as suggested. (Line 225)

L189, recognized and internalized within MVB in intraluminal vesicles

              We rewrote this as requested. (Lines 227-228)

MVB abbreviation should be explained only once, when first appears.

              The second explanation of MVB abbreviation, which was in the legend of Figure 1, has been removed from the revised ms.

L192, to increase activity

              This phrase has now been changed. (Line 230)

Chapter 3

L205, (TXNIP) belongs to. Simpler and less words is better

              This has been rendered more concisely. (Line 243)

L207 in the plasma membrane

              Done. (Line 245)

L209 clathrin coated endocytic vesicles.

              Done. (Line 248)

L216. Too long sentence, 6 lines, split.

              In the revised ms, the indicated sentence has been split into two. It now reads, “In mammalian cells stimulated by growth factors, in which energy demand is supposedly raised for cell growth and division, AKT may increase the glucose uptake rate by inhibiting internalization of hexose transporters as well as by promoting translocation of GLUT4 from cytoplasmic storage vesicles to the plasma membrane [8,37]. On the other hand, in fission yeast cells under nitrogen-rich conditions, Gad8/AKT may block unnecessary protein degradation for amino acid replenishment.” (Lines 254-260)

L224, both result in phosphorylation of.. serine 308 residue. Simpler and shorter is better.

              We agree. (Line 263)

Concluding remarks

L232, to the endocytic vesicles

              OK. (Line 272)

L234, remains to be determined

              Edited accordingly. (Line 274)

References

Be sure that Schizosaccharomyces pombe is in italics.

              “Schizosaccharomyces pombe” and “S. pombe” are italicized throughout the revised reference section.

Round 2

Reviewer 1 Report

This is now a useful contribution

Reviewer 2 Report

The manuscript is now suitable for publication.